# DTC-WSI: Dynamic Token Compression for Whole-Slide Images

**Tawsifur Rahman**[1]                                                        ARAHMA34@JHU.EDU
[1] *Biomedical Engineering, Johns Hopkins University*

**Aliasghar Tarkhan**[2]                                       TARKHAN.ALIASGHAR@GMAIL.COM
[2] *Johnson & Johnson, MedTech*

**Rama Chellappa**[3]                                                        RCHELLA4@JHU.EDU
[3] *Johns Hopkins University*

**Alexander S. Baras**[4]                                                      BARAS@JHMI.EDU
[4] *School of Medicine, Johns Hopkins University*

**Editors:** Accepted for publication at MIDL 2026

## Abstract

Whole-slide images (WSIs) contain tens of thousands of heterogeneous patches, making transformer-based multiple-instance learning (MIL) computationally expensive due to quadratic attention costs and substantial redundancy in tissue morphology. Existing token-reduction approaches for WSI analysis rely primarily on pruning, which discards information early in training and destabilizes optimization under weak supervision. We propose **Dynamic Token Compression for Whole-Slide Images (DTC-WSI)**, a token-efficient MIL framework that performs *progressive, importance-aware* WSI compression. DTC-WSI integrates a lightweight saliency network with a multi-stage token compressor that combines *bipartite similarity matching* and *soft differentiable pruning* to gradually eliminate redundant or non-diagnostic patches. During training, soft gates enable stable gradient flow, while inference employs deterministic compression for substantial acceleration. This curriculum-style compression preserves discriminative morphology and dramatically reduces computational burden. Across four WSI benchmarks (TCGA-NSCLC, TCGA-BRCA, TCGA-RCC, PANDA), DTC-WSI achieves **5−10× token reduction**, **up to 5.3× faster inference**, and **20–40% lower memory usage**, while improving MIL classification accuracy by **2–4%** over state-of-the-art baselines. Our results demonstrate that dynamic token compression is a powerful and scalable alternative to pruning, enabling efficient transformer-based WSI analysis while improving accuracy.

**Keywords:** Computational pathology, Token merging, Dynamic token pruning, Weakly supervised learning

## 1. Introduction

Whole-slide images (WSIs) are gigapixel-scale pathology scans that exhibit rich and highly heterogeneous morphological patterns over extremely large spatial extents (El Nahhas et al., 2025; Srinidhi et al., 2021; Cui and Zhang, 2021). Since WSIs cannot be processed at native resolution, modern computational pathology pipelines partition each slide into thousands of fixed-size patches and employ multiple instance learning (MIL) to aggregate patch-level representations into slide-level predictions (Obeid et al., 2025; Gurcan et al., 2009; Brunyé

et al., 2010). Recent attention-based and transformer-based MIL models—including AB-MIL (Ilse et al., 2018), CLAM (Lu et al., 2021), TransMIL (Shao et al., 2021), DSMIL (Li et al., 2021), as well as hierarchical architectures such as HIPT (Chen et al., 2022)—have demonstrated strong performance across tasks such as cancer subtyping, grading, and prognosis. However, these approaches face a fundamental scalability challenge: a single diagnostic WSI can yield tens of thousands of patch tokens, leading to substantial computational and memory overhead in MIL scoring modules and transformer attention layers (Rahman et al., 2025a; Yang et al., 2022; Kapse et al., 2024).

This challenge is exacerbated by the structural properties of histopathology images. Large regions contain visually redundant or weakly discriminative tissue—including stroma, adipose, necrosis, and repeated tumor textures (Tang et al., 2024). Treating all patches as independent tokens forces models to process extensive redundancy, increasing computation without adding discriminative signal (Rahman et al., 2025b). Prior attempts to mitigate this include hierarchical MIL (Yue et al., 2025), patch clustering (Sharma et al., 2021), and token pruning (Tang et al., 2023a; Rao et al., 2021). Yet pruning irreversibly discards tokens and risks removing diagnostically relevant regions, a severe limitation under weak supervision where slide-level labels provide no guidance for early-stage pruning decisions (Jiang et al., 2023; Shi et al., 2021; Lyu et al., 2025).

Meanwhile, the natural-image community has demonstrated that *token merging* can accelerate Vision Transformers by fusing redundant tokens rather than removing them. Methods such as ToMe (Bolya and Hoffman, 2023) exploit similarity structure to merge tokens without losing information. However, these methods have not been adapted to computational pathology, where redundancy patterns are more complex, token counts are orders of magnitude larger, and merging must be guided by task-driven saliency to avoid collapsing diagnostically meaningful structures (Zhen et al., 2025; Hu et al., 2024; Wu et al., 2025).

To address these limitations, we propose **Dynamic Token Compression for Whole-Slide Images (DTC-WSI)**, a unified framework that combines *similarity-guided token merging* with *importance-guided pruning* in a progressive multi-stage pipeline. DTC-WSI fuses redundant patches via efficient bipartite matching while learning patch saliency through a differentiable importance network trained with slide-level supervision. Unlike single-step or merge-only approaches, our curriculum-style multi-stage compression gradually reduces tokens, preventing early information collapse and stabilizing saliency estimation. This hybrid design enables efficient gigapixel WSI processing while preserving diagnostically critical regions. Our contributions are summarized as follows:

1. **A unified multi-stage token compression framework** that jointly performs similarity-guided token merging and importance-guided pruning, enabling aggressive token reduction while preserving diagnostic morphology.

2. **A differentiable importance network** that learns patch saliency under weak supervision, guiding compression during training and enabling deterministic, high-efficiency inference.

3. **Comprehensive evaluation on four major WSI benchmarks** (TCGA-NSCLC, TCGA-BRCA, TCGA-RCC, PANDA), demonstrating that DTC-WSI achieves **5–10× token reduction**, **up to 5.3× faster inference**, **20–40% lower memory**

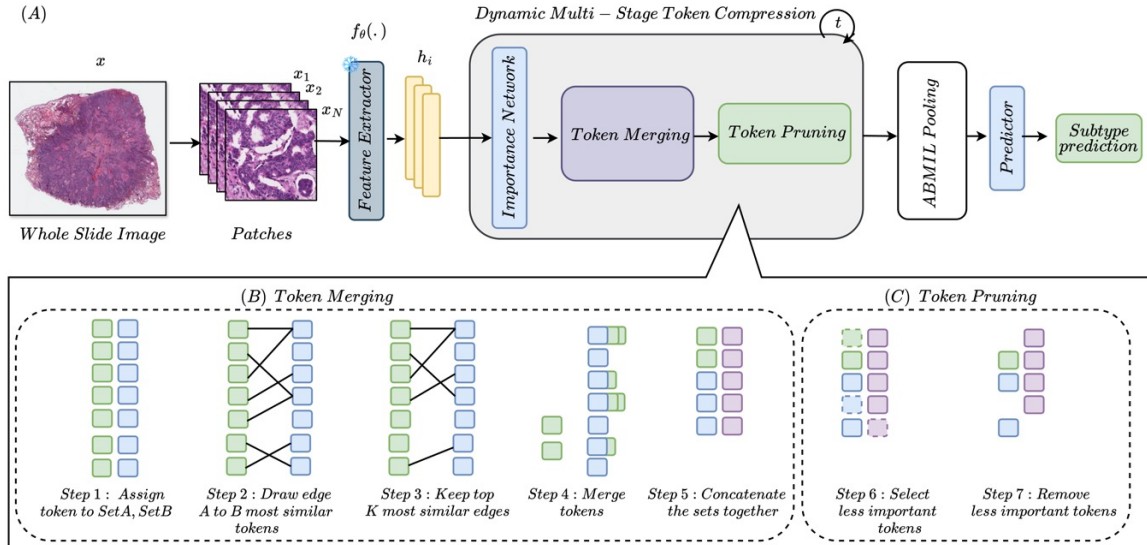

Figure 1: Overview of the proposed **Dynamic Token Compression (DTC-WSI)** framework. **(A)** End-to-end pipeline: patch extraction, feature encoding, multi-stage token compression, and MIL prediction. **(B)** Token merging: similar patches are fused into unified representations via bipartite soft matching. **(C)** Token pruning: low-importance tokens are removed to produce a compact, discriminative set for classification.

**usage**, and **2–4% accuracy gains** over state-of-the-art MIL and token-efficient baselines.

## 2. Methods

### 2.1. Overview

Whole-slide images (WSIs) contain tens of thousands of patches, making conventional MIL and transformer models computationally prohibitive due to quadratic attention and high memory demands. To address this, we propose **Dynamic Token Compression for Whole-Slide Images (DTC-WSI)**, a framework that *learns* to compress WSI patch embeddings in a task-aware manner. DTC-WSI progressively reduces tokens across multiple stages by combining (1) **similarity-guided merging** to fuse redundant patches and (2) **importance-guided pruning** to discard low-saliency regions. Compression is applied *softly* during training, enabling the importance network to learn reliable saliency estimates, and *deterministically* at inference for fast, scalable deployment. The final compact token set is aggregated using attention-based MIL to produce slide-level predictions.

## 2.2. Patch Extraction and Feature Encoding

A whole-slide image (WSI) is denoted by $x$, which is tiled into $N$ non-overlapping patches $\{x_1, x_2, \ldots, x_N\}$ after tissue detection and background removal. Each patch is processed by a pretrained encoder (CONCH (Lu et al., 2024)) to obtain a semantic feature embedding:

$$h_i^{(0)} = f_\theta(x_i) \in \mathbb{R}^D, \qquad i = 1, \ldots, N. \tag{1}$$

To preserve spatial information, optional positional embeddings $p_i$ concatenated with the visual features:

$$\tilde{h}_i^{(0)} = [h_i^{(0)} \,\|\, p_i].$$

forming the initial token matrix $H^{(0)} = [\tilde{h}_1^{(0)}, \ldots, \tilde{h}_N^{(0)}]^\top \in \mathbb{R}^{N \times (D+1)}$.

All supervision is provided at the slide level; no patch-level labels are used during training.

## 2.3. Importance Network

WSIs contain large regions of redundant or clinically irrelevant tissue, making it essential to quantify which patch embeddings contribute meaningfully to the slide prediction. The **Importance Network** $g_\phi$ assigns a saliency score to each token at stage t:

$$s_i^{(t)} = g_\phi(\tilde{h}_i^{(t-1)}), \tag{2}$$

where $g_\phi$ is a two-layer MLP with GELU activation. Scores are normalized into importance weights:

$$\alpha_i^{(t)} = \frac{\exp(s_i^{(t)})}{\sum_{j=1}^{N^{(t)}} \exp(s_j^{(t)})},$$

which induces soft competition among tokens. Early in training, the distribution remains diffuse; as learning progresses, high-saliency tumor regions receive larger weights.

## 2.4. Dynamic Multi-Stage Token Compression

To prevent catastrophic loss of diagnostic evidence, we adopt a **multi-stage compression** schedule:

$$N^{(0)} = N \;\rightarrow\; N^{(1)} \;\rightarrow\; N^{(2)} \;\rightarrow\; N^{(3)}, \tag{3}$$

where each $N_{t+1}$ is determined by a retention ratio $r : N^{(t+1)} = r \cdot N^{(t)}$

The number of token merges required in stage $t$ is: $K^{(t)} = N^{(t)} - N^{(t+1)}$

Each stage consists of: 1) **Bipartite soft matching for token fusion**, and 2) **Importance-guided pruning**.

### 2.4.1. BIPARTITE SOFT MATCHING FOR TOKEN FUSION

To avoid the $O(N^2)$ complexity of full similarity search, tokens in stage $t$ are partitioned into alternating subsets:

$$A = [\tilde{h}_1^{(t-1)}, \tilde{h}_3^{(t-1)}, \tilde{h}_5^{(t-1)}, \dots], \quad B = [\tilde{h}_2^{(t-1)}, \tilde{h}_4^{(t-1)}, \tilde{h}_6^{(t-1)}, \dots].$$

For each aligned pair $(i, j)$, cosine similarity is computed:

$$\mathrm{sim}(i, j) = \frac{\langle \tilde{h}_i^{(t-1)}, \tilde{h}_j^{(t-1)} \rangle}{\|\tilde{h}_i^{(t-1)}\| \, \|\tilde{h}_j^{(t-1)}\|}.$$

A merge utility incorporating importance consistency is defined as:

$$u_{ij}^{(t-1)} = \lambda \, \mathrm{sim}(i, j) - (1 - \lambda) \, |\alpha_i^{(t)} - \alpha_j^{(t)}|.$$

The Top-$K^{(t)}$ pairs are fused using importance-weighted averaging:

$$\tilde{h}_l^{(t)} = \frac{\alpha_i^{(t)} \tilde{h}_i^{(t-1)} + \alpha_j^{(t)} \tilde{h}_j^{(t-1)}}{\alpha_i^{(t)} + \alpha_j^{(t)}}. \tag{4}$$

### 2.4.2. IMPORTANCE-GUIDED TOKEN PRUNING

After token merging, low-saliency tokens are suppressed. During training, pruning is differentiable:

$$m_l^{(t)} = \sigma\left(\gamma(\alpha_l^{(t)} - \tau)\right), \quad \tilde{h}_l^{(t)} = m_l^{(t)} \, \tilde{h}_l^{(t)}.$$

During inference, deterministic Top-$N^{(t)}$ pruning is applied:

$$H^{(t)} = \mathrm{TopK}\left(H^{(t-1)}, \alpha^{(t-1)}, N^{(t-1)}\right).$$

## 2.5. MIL Aggregation and Prediction

After $t$ compression stages, the final tokens $H^{(t)}$ are passed to an attention-based MIL module. Attention weights:

$$a_i = \frac{\exp(w^\top \tanh(W h_i^{(t)}))}{\sum_{j=1}^{M} \exp(w^\top \tanh(W h_j^{(t)}))}.$$

The final slide-level representation is computed as a weighted sum of the compressed tokens, $z = \sum_{i=1}^{N^{(t)}} a_i \tilde{h}_i^{(t)}$, where the attention weights emphasize diagnostically informative regions. This embedding is then passed through a linear classifier followed by a softmax layer to produce the slide-level prediction, $\hat{y} = \mathrm{softmax}(W_c z + b_c)$.

## 2.6. Loss Function

We supervise slide-level predictions using cross-entropy, $\mathcal{L}_{\mathrm{cls}} = \mathrm{CE}(y, \hat{y})$, and encourage the importance network to assign sparse, selective saliency through an $\ell_1$ regularizer, $\mathcal{L}_{\mathrm{sparse}} = \beta \sum_{t=0}^{T} \|\alpha^{(t)}\|_1$. The full training objective combines both terms to promote discriminative yet compact representations, the composite loss is given as:

$$\mathcal{L} = \mathcal{L}_{\mathrm{cls}} + \mathcal{L}_{\mathrm{sparse}}.$$

A complete step-by-step description of the algorithm is provided in **Appendix D**.

## 3. Results

### 3.1. Datasets

We evaluated DTC-WSI across four large-scale histopathology classification benchmarks and one cellular-level morphology task to assess both its robustness on diverse cancer subtyping problems and its generalizability beyond WSIs.

**TCGA-NSCLC.** (Tomczak et al., 2015) This dataset comprises 993 whole-slide images (WSIs) from Formalin-Fixed Paraffin-Embedded (FFPE) tissue samples, with 507 slides corresponding to lung adenocarcinoma (LUAD) and 486 to lung squamous cell carcinoma (LUSC).

**TCGA-BRCA.** (Tomczak et al., 2015) The TCGA-BRCA dataset includes 938 FFPE WSIs, of which 772 are diagnosed with Invasive Ductal Carcinoma (IDC) and 166 with Invasive Lobular Carcinoma (ILC).

**TCGA-RCC.** (Tomczak et al., 2015) The TCGA-RCC cohort contains 884 diagnostic WSIs covering three renal cell carcinoma subtypes: Chromophobe (TCGA-KICH), Clear Cell (TCGA-KIRC), and Papillary (TCGA-KIRP). The dataset includes 111 slides from 99 CRCC cases, 489 slides from 483 CCRCC cases, and 284 slides from 264 PRCC cases. On average, each slide contributed approximately 13,900 patches at $\times 20$ magnification.

**PANDA.** (Bulten et al., 2022) The PANDA dataset consists of 12,625 prostate biopsy WSIs collected from six different institutions. The dataset includes 3,628 non-tissue/background slides, 3,151 non-epithelium/non-cancerous slides, 1,644 benign slides, and 4,202 cancerous slides. For our classification task, we focused on benign and cancerous slides to ensure a clinically meaningful evaluation.

### 3.2. Experimental Setup and Evaluation Metrics

All experiments were implemented in PyTorch and executed on a compute server equipped with four NVIDIA Tesla V100 GPUs and 32 CPU cores. Models were trained using a batch size of 256, the Adam optimizer with an initial learning rate of 0.001, and early stopping based on validation performance. We employed **5-fold cross-validation** for all datasets to ensure robust performance estimation. The token retention ratio $r$ was used to tune the effective thresholds for both similarity-based merging and importance-guided pruning, with hyperparameters (merge utility weights, pruning ratios, and sparsity coefficient) optimized separately for each dataset. We report classification performance using Accuracy and Area Under the ROC Curve (AUC), where multi-class accuracy is computed as the average per-class accuracy and AUC is macro-averaged across classes.

Table 1: Comparison of **DTC-WSI** ($r = 0.4$) with MIL and token-efficient baselines across four datasets. Results are reported as mean $\pm$ std over 5 folds. All methods use the same pretrained encoder (CONCH (Lu et al., 2024)) and identical hardware settings.

| Model | TCGA-NSCLC | | TCGA-BRCA | | TCGA-RCC | | PANDA | |
|---|---|---|---|---|---|---|---|---|
| | Acc | AUC | Acc | AUC | Acc | AUC | Acc | AUC |
| ABMIL (Ilse et al., 2018) | 94.7±0.6 | 95.4±0.5 | 93.4±0.7 | 94.1±0.6 | 92.6±0.6 | 93.5±0.5 | 91.2±0.7 | 92.1±0.6 |
| CLAM-MB (Lu et al., 2021) | 95.8±0.5 | 96.6±0.4 | 94.5±0.6 | 95.3±0.5 | 93.9±0.6 | 94.8±0.5 | 92.4±0.6 | 93.4±0.5 |
| DSMIL (Li et al., 2021) | 95.3±0.6 | 96.1±0.5 | 94.0±0.6 | 94.8±0.6 | 93.4±0.6 | 94.2±0.5 | 91.9±0.7 | 92.9±0.6 |
| TransMIL (Shao et al., 2021) | 96.4±0.5 | 97.2±0.4 | 95.6±0.5 | 96.4±0.5 | 94.8±0.5 | 95.6±0.4 | 92.8±0.6 | 93.7±0.5 |
| HIPT (Chen et al., 2022) | 96.2±0.5 | 97.0±0.4 | 95.3±0.6 | 96.1±0.5 | 94.6±0.5 | 95.4±0.5 | 92.6±0.6 | 93.6±0.5 |
| PANTHER (Song et al., 2024) | 96.8±0.4 | 97.6±0.4 | 96.0±0.5 | 96.8±0.4 | 95.2±0.5 | 96.1±0.4 | 93.3±0.5 | 94.2±0.4 |
| SPT (Hou et al., 2024) | 95.9±0.5 | 96.7±0.4 | 94.8±0.6 | 95.6±0.5 | 93.8±0.6 | 94.6±0.5 | 92.3±0.6 | 93.2±0.5 |
| ToMe (Bolya and Hoffman, 2023) | 91.0±0.8 | 91.8±0.7 | 90.0±0.8 | 90.8±0.7 | 89.1±0.8 | 90.0±0.7 | 87.5±0.9 | 88.4±0.8 |
| PatchGD (Gupta et al., 2023) | 94.2±0.6 | 95.0±0.5 | 93.2±0.7 | 94.0±0.6 | 92.4±0.6 | 93.3±0.6 | 90.7±0.7 | 91.6±0.6 |
| MHIM-MIL (Tang et al., 2023b) | 93.5±0.7 | 94.3±0.6 | 92.8±0.7 | 93.6±0.6 | 91.9±0.7 | 92.8±0.6 | 90.1±0.8 | 91.0±0.7 |
| Longformer (Zhang et al., 2021) | 92.1±0.8 | 92.9±0.7 | 91.4±0.8 | 91.2±0.7 | 89.6±0.8 | 90.4±0.7 | 88.9±0.8 | 89.8±0.7 |
| 2DMambaMIL (Zhang et al., 2025) | 94.3±0.6 | 95.1±0.5 | 91.7±0.7 | 92.5±0.6 | 90.7±0.7 | 91.6±0.6 | 89.0±0.8 | 90.9±0.7 |
| Random Sampling | 78.5±1.1 | 79.1±1.0 | 72.2±1.2 | 73.4±1.1 | 81.4±1.0 | 81.7±0.9 | 71.6±1.3 | 72.1±1.2 |
| **DTC-WSI (Ours)** | **98.3±0.3** | **98.9±0.2** | **97.4±0.3** | **97.9±0.3** | **96.8±0.4** | **97.5±0.3** | **94.8±0.4** | **95.6±0.3** |

### 3.3. Performance Comparison

We evaluated DTC-WSI on four benchmark WSI datasets—TCGA-NSCLC, TCGA-BRCA, TCGA-RCC, and PANDA—and compared it against a comprehensive set of state-of-the-art MIL and token-efficient approaches. These include classical MIL models (ABMIL, CLAM-MB, DSMIL), transformer-based and hierarchical methods (TransMIL, HIPT, PANTHER, SPT), as well as recent efficiency-oriented baselines such as ToMe, PatchGD, MHIM-MIL, Longformer, and 2DMambaMIL. For fair comparison, **all methods use the same pretrained feature encoder** ( CONCH (Lu et al., 2024)). In addition, we include a **random sampling baseline** that retains the same fraction of tokens ($r = 0.4$) as DTC-WSI to isolate the effect of *dynamic* compression from simple token reduction. The results are summarized in Table 1.

DTC-WSI consistently achieves the best performance across all datasets, reaching **98.3%** accuracy on TCGA-NSCLC, **97.4%** on TCGA-BRCA, **96.8%** on TCGA-RCC, and **94.8%** on PANDA. Compared to strong MIL and token-efficient baselines, DTC-WSI improves accuracy by approximately **1.8–3.6%** and AUC by **1.6–3.5%** across datasets, while retaining only **40%** of the original tokens.

Importantly, random sampling—despite using the same token budget—exhibits a substantial performance drop across all datasets, indicating that efficiency gains alone do not account for the improvements. This underscores the importance of *saliency-aware dynamic compression*: DTC-WSI preserves diagnostically informative regions via importance-guided pruning and similarity-aware merging, rather than indiscriminate token removal. We find that $r = 0.4$ provides the best accuracy–efficiency trade-off (Appendix B). Overall, DTC-WSI achieves a superior balance compared to static sampling and existing token-efficient methods.

Table 2: Computational efficiency and accuracy of **DTC-WSI** under different token reten-
tion ratios. All rows correspond to the same DTC-WSI framework with different
final token budgets.

| Final Retention | Acc (%) | FLOPs (G) | MIL Aggregation Only | | Full Pipeline | | Speedup |
|---|---|---|---|---|---|---|---|
| | | | GPU Mem (GB) | Time (ms) | GPU Mem (GB) | Time (ms) | |
| r = 1.0 (no compression) | 94.6 | 118.4 | 3.1 | 420 | 14.2 | 2150 | 1.0× |
| r = 0.7 (Light Compression) | 96.1 | 62.7 | 2.2 | 260 | 10.3 | 1190 | 1.8× |
| r = 0.5 (Moderate Compression) | 97.1 | 38.4 | 1.6 | 170 | 8.1 | 720 | 3.0× |
| **r = 0.4 (Best)** | **98.3** | **24.3** | **1.1** | **95** | **6.4** | **410** | **5.3×** |

Table 3: **Training, Inference Efficiency, and Accuracy Comparison (TCGA-
NSCLC).** All methods use the same pretrained encoder ( CONCH (Lu et al.,
2024)) and identical hardware settings.

| Method | Acc (%) | Train Time / Epoch (s) | Peak GPU Memory (GB) | Inference Time / WSI (ms) |
|---|---|---|---|---|
| ABMIL | 94.7 | 312 | 14.2 | 1860 |
| TransMIL | 96.4 | 428 | 18.9 | 2740 |
| ToMe | 91.0 | 228 | 9.4 | 870 |
| MHIM-MIL | 93.5 | 214 | 8.9 | 890 |
| Random Sampling ($r$=0.4) | 78.5 | 205 | 8.6 | 330 |
| **DTC-WSI (Ours, $r$=0.4)** | **98.3** | **236** | **9.1** | **410** |

### 3.4. Computational Efficiency of Token Compression

Beyond accuracy improvements, DTC-WSI provides substantial computational savings through
progressive multi-stage token compression. Table 2 reports the impact of different token
retention ratios within the same DTC-WSI framework on both efficiency and accuracy.
As the retention ratio decreases, FLOPs, GPU memory usage, and inference latency con-
sistently decline, while classification accuracy steadily improves. Using the full token set
($r = 1.0$), DTC-WSI requires 118.4 G FLOPs, 14.2 GB GPU memory, and 2150 ms per
WSI. Light compression ($r = 0.7$) nearly halves computation and yields a 1.8× speedup,
while moderate compression ($r = 0.5$) further reduces inference time to 720 ms.

The best trade-off is achieved at $r = 0.4$, where DTC-WSI retains only **40%** of the
original tokens yet attains the highest accuracy (98.3% on TCGA-NSCLC). At this setting,
FLOPs are reduced to 24.3 G, peak GPU memory drops to 6.4 GB, and end-to-end infer-
ence time decreases to 410 ms, corresponding to a **5.3× speedup** over the uncompressed
setting. Importantly, these efficiency gains are accompanied by improved predictive per-
formance, indicating that dynamic compression removes redundant and low-saliency tokens
while preserving diagnostically relevant regions.

To contextualize these gains, Table 3 also reports training and inference efficiency com-
parisons against standard MIL and token-efficient baselines. We include a **random sam-
pling baseline that retains the same 40% token budget** as DTC-WSI. Although
random sampling achieves low inference latency due to aggressive token reduction, it leads
to severe accuracy degradation (Table 1), highlighting that computational savings alone
are insufficient. In contrast, DTC-WSI maintains competitive training cost and memory
usage while significantly outperforming random and static selection strategies in accuracy,
demonstrating the necessity of *saliency-aware, dynamic token allocation*. Overall, these

results confirm that DTC-WSI delivers a superior accuracy–efficiency trade-off compared to both naïve sampling and existing token-efficient MIL approaches.

Table 4: Ablation study comparing merging and pruning strategies (r=0.4) across four datasets. Metrics reported as Accuracy (Acc) and AUC (%).

| Model Variant | TCGA-NSCLC | | TCGA-BRCA | | TCGA-RCC | | PANDA | |
|---|---|---|---|---|---|---|---|---|
| | Acc | AUC | Acc | AUC | Acc | AUC | Acc | AUC |
| Random Merging | 95.0 | 96.1 | 94.0 | 95.1 | 93.1 | 94.3 | 91.6 | 92.8 |
| Random Pruning | 94.6 | 95.7 | 93.6 | 94.8 | 92.9 | 94.1 | 91.2 | 92.4 |
| Only Merge (Similarity-Guided) | 96.4 | 97.6 | 95.3 | 96.3 | 94.5 | 95.7 | 93.0 | 94.2 |
| Only Prune (Importance-Guided) | 95.7 | 96.9 | 94.7 | 95.8 | 93.9 | 95.1 | 92.4 | 93.5 |
| **Ours (Dynamic Merge + Prune)** | **98.3** | **98.9** | **97.4** | **97.9** | **96.8** | **97.5** | **94.8** | **95.6** |

Table 5: Sensitivity analysis of DTC-WSI to multi-stage retention schedules. Final token budget is fixed to ~40% across all settings.

| Schedule | Stage 1 ($r_1$) | Stage 2 ($r_2$) | Stage 3 ($r_3$) | TCGA-NSCLC | | TCGA-BRCA | | TCGA-RCC | | PANDA | |
|---|---|---|---|---|---|---|---|---|---|---|---|
| | | | | Acc | AUC | Acc | AUC | Acc | AUC | Acc | AUC |
| Uniform (default) | 0.74 | 0.74 | 0.74 | 98.3 | 98.9 | 97.4 | 97.9 | 96.8 | 97.5 | 94.8 | 95.6 |
| Balanced | 0.70 | 0.70 | 0.82 | 98.2 | 98.8 | 97.3 | 97.8 | 96.7 | 97.4 | 94.6 | 95.4 |
| Early-aggressive | 0.55 | 0.80 | 0.91 | 97.6 | 98.1 | 96.8 | 97.2 | 96.1 | 96.8 | 94.0 | 94.7 |
| Late-aggressive | 0.85 | 0.70 | 0.67 | 98.0 | 98.6 | 97.1 | 97.6 | 96.5 | 97.2 | 94.4 | 95.1 |
| 2-stage only | 0.63 | 0.63 | – | 97.8 | 98.4 | 96.9 | 97.4 | 96.3 | 97.0 | 94.2 | 94.9 |

## 3.5. Ablation Studies

**Ablation on Merging and Pruning Strategies.** Table 4 presents an ablation study isolating the effects of token merging and pruning strategies under a fixed token budget of $r = 0.4$ for all variants. Random merging and random pruning result in noticeable performance degradation across all datasets, indicating that indiscriminate compression disrupts discriminative slide-level signals. Using similarity-guided merging alone consistently outperforms random merging by preserving redundant yet morphologically coherent regions, while importance-guided pruning alone yields moderate gains by suppressing low-saliency tokens. However, neither strategy alone matches the full DTC-WSI framework. Combining similarity-guided merging with importance-guided pruning achieves the best performance across all benchmarks, improving accuracy by 2–4% over single-strategy variants. These results demonstrate that dynamic, saliency-aware merging and pruning are both necessary and complementary, and that performance gains cannot be attributed to token reduction alone, but to the proposed joint dynamic compression mechanism.

**Sensitivity to Multi-Stage Retention Schedules.** Table 5 analyzes the sensitivity of DTC-WSI to different multi-stage token retention schedules, while fixing the final token budget to approximately 40% across all settings. We observe that performance remains consistently strong across a wide range of stage-wise retention ratios, indicating that DTC-WSI is not overly sensitive to precise hyperparameter choices. The uniform schedule ($r_1 = r_2 = r_3 = 0.74$) yields the best overall performance and is used as the default configuration.

More aggressive early or late compression leads to a modest accuracy drop, suggesting that progressively reducing tokens helps preserve diagnostically relevant regions under weak supervision. Overall, these results demonstrate that the proposed multi-stage compression strategy is stable, robust, and does not require fine-tuning of stage-wise retention ratios to achieve strong performance.

**Encoder and backbone ablation.** Table 6 analyzes the impact of encoder choice and MIL backbone on predictive performance and computational efficiency. Across both encoders (CONCH and Virchow2) and MIL backbones (ABMIL and TransMIL), DTC-WSI consistently improves accuracy while substantially reducing GPU memory usage and inference time. For instance, with the CONCH encoder, DTC-ABMIL improves accuracy from 94.7% to 98.3% while reducing GPU memory by more than 2× and inference time by over 5×. Similar trends are observed for transformer-based aggregation, where DTC-TransMIL outperforms vanilla TransMIL while reducing memory consumption and runtime by approximately 3–5×. Importantly, these gains are consistent across encoders of different strengths, indicating that the improvements stem from the proposed dynamic token compression rather than the upstream feature extractor. Overall, the results demonstrate that DTC-WSI is both encoder-agnostic and backbone-agnostic, providing a favorable accuracy–efficiency trade-off for both lightweight and transformer-based MIL pipelines.

Table 6: Ablation of encoder choice and MIL backbone. All models are evaluated under identical data splits and training protocols.

| Model | Encoder | Tokens Retained | Acc (%) | GPU Mem (GB) | Inference Time (ms/WSI) |
|---|---|---|---|---|---|
| ABMIL | CONCH | 1.0 | 94.7 | 14.2 | 2150 |
| DTC-ABMIL | CONCH | 0.4 | **98.3** | **6.4** | **410** |
| TransMIL | CONCH | 1.0 | 95.2 | 18.9 | 2740 |
| DTC-TransMIL | CONCH | 0.4 | **98.9** | **7.8** | **480** |
| ABMIL | Virchow2 | 1.0 | 92.1 | 14.5 | 1810 |
| DTC-ABMIL | Virchow2 | 0.4 | **96.0** | **6.6** | **350** |
| TransMIL | Virchow2 | 1.0 | 94.3 | 19.1 | 2210 |
| DTC-TransMIL | Virchow2 | 0.4 | **97.3** | **8.1** | **430** |

**Ablation on Sparsity Regularization.** We analyze the effect of the sparsity regularizer $\mathcal{L}_{\text{sparse}}$ applied to the importance scores by training DTC-WSI with and without the $\ell_1$ penalty. Removing $\mathcal{L}_{\text{sparse}}$ results in denser importance distributions, which weakens the pruning behavior and leads to higher token retention and increased inference cost. This also causes a consistent degradation in classification performance. In contrast, incorporating the sparsity term encourages selective saliency assignment, stabilizes multi-stage token compression, and yields both improved accuracy and computational efficiency. These results demonstrate that $\mathcal{L}_{\text{sparse}}$ is a critical component for learning compact yet discriminative representations under weak slide-level supervision.

## 4. Visualization of Token Compression

Figure 2 illustrates the full multi-stage compression process performed by DTC-WSI. Panel (A) shows the original WSI along with overlaid heatmaps depicting the model output after

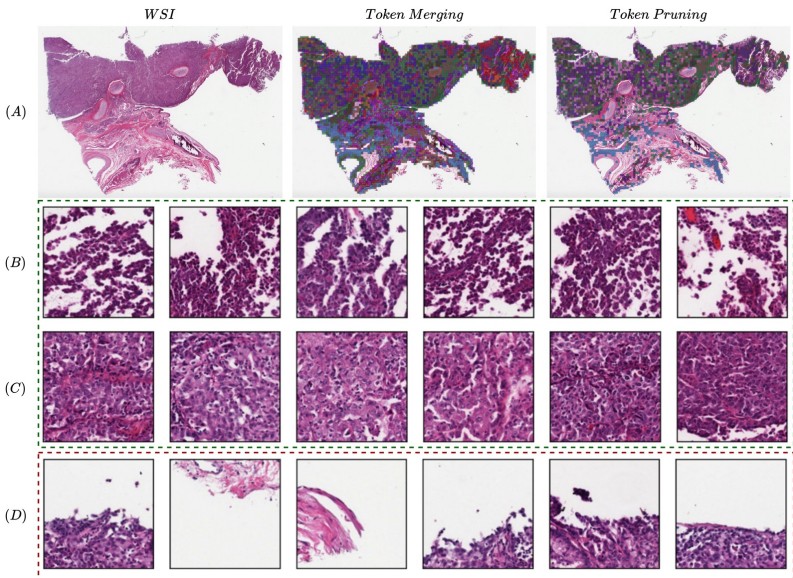

Figure 2: Visualization of multi-stage token compression in DTC-WSI: (A) Original WSI with post-merging and post-pruning heatmaps. (B–C) Similar patches merged into unified tokens (green), and (D) low-saliency patches removed by pruning (red).

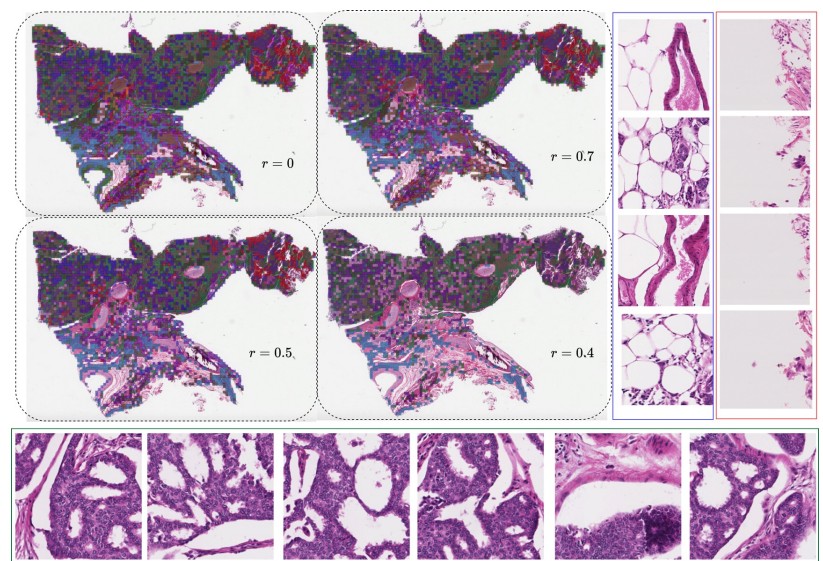

Figure 3: Visualization of multi-stage token compression in DTC-WSI across retention ratios $r \in 1.0, , 0.7, , 0.5, , 0.4$. Example merged patches (e.g., adipose or stroma) are shown in blue boxes, pruned patches (e.g., background or slide borders) in red boxes, and high-importance patches retained for final prediction (e.g., tumor regions) in green boxes, highlighting diagnostically relevant tissue patterns.

Table 7: Ablation study on sparsity regularization. Effect of removing the $\ell_1$ sparsity loss $\mathcal{L}_{\text{sparse}}$ on performance and compression behavior (TCGA-NSCLC).

| Setting | Acc (%) | AUC (%) | Tokens Retained | Inference Time (ms) |
|---|---|---|---|---|
| DTC-WSI w/o $\mathcal{L}_{\text{sparse}}$ | 96.9 | 97.6 | 0.52 | 610 |
| DTC-WSI (full, Ours) | **98.3** | **98.9** | **0.40** | **410** |

similarity-guided token merging and after importance-guided pruning. Panels (B) and (C) present examples of visually similar patches that are merged into unified representations; these merged groups are highlighted with green borders, demonstrating how redundant regions—such as uniform stromal areas or repeated tumor patterns—are effectively consolidated. Panel (D) displays patches removed through importance-guided pruning, marked with red borders, revealing low-saliency regions that contribute minimally to the slide-level prediction. Overall, these visualizations show that DTC-WSI performs structured, interpretable compression: reducing redundancy through merging while selectively pruning non-informative regions, ultimately preserving the most diagnostically meaningful tissue patterns.

Figure 3 illustrates how DTC-WSI progressively compresses a whole-slide image as the token retention ratio decreases from $r = 1.0$ (no compression) to $r = 0.7$, $r = 0.5$, and $r = 0.4$. For each retention level, we visualize the WSI after applying similarity-guided token merging and importance-guided pruning. Each panel illustrates the effect of these operations as the token budget decreases. Example merged patches (blue boxes) correspond to visually homogeneous and redundant regions (e.g., adipose or stromal tissue), which are fused to reduce redundancy. Pruned patches (red boxes) highlight low-saliency or non-informative regions (e.g., slide borders or artifacts) that are removed during compression. In contrast, high-importance patches retained for final prediction (green boxes) capture diagnostically relevant tumor tissue patterns. At $r = 1.0$, all extracted patches are preserved, resulting in a dense and highly redundant representation, whereas by $r = 0.4$ the representation becomes substantially more compact while preserving salient tissue structures. These visualizations demonstrate that DTC-WSI performs semantically meaningful compression by preserving informative regions while aggressively removing redundancy, thereby concentrating model capacity on morphologically relevant content and enabling both computational efficiency and improved predictive performance.

## 5. Conclusion

We presented **DTC-WSI**, a scalable framework for token-efficient whole-slide image analysis. By combining similarity-guided merging with importance-guided pruning in a progressive multi-stage pipeline, DTC-WSI removes redundancy while preserving diagnostically essential information. The method supports differentiable compression during training and deterministic reduction at inference, achieving **5–10× token reduction**, **5.3× faster inference**, and **40% lower memory usage** without sacrificing accuracy. Across four benchmark datasets, DTC-WSI improves classification performance by **2–4%**, demonstrating that compression can enhance representation quality rather than degrade it.

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

## Appendix A.

## Whole Slide Image Preprocessing

Whole slide image (WSI) preprocessing begins with automated tissue segmentation. Each WSI is first loaded into memory at a downsampled resolution, such as $20\times$, and converted from RGB to HSV colorspace. Tissue regions (foreground) are identified by thresholding the saturation channel after applying median blurring to smooth edges. A binary mask is then generated and refined using morphological closing to eliminate small gaps and holes. The contours of detected tissue regions are filtered based on an area threshold, ensuring only relevant regions are retained for further processing. The segmentation mask for each slide is also available for optional visual inspection. To facilitate manual adjustments, a human-readable text file is generated, listing processed files along with editable segmentation parameters. Once segmentation is complete, $256\times256$ patches are extracted from within the segmented contours at the specified magnification. These patches, along with their coordinates and slide metadata, are stored in the HDF5 hierarchical data format. The number of extracted patches per slide varies significantly—ranging from hundreds in biopsy slides at $20\times$ magnification to hundreds of thousands in large resection slides at $40\times$ magnification.

## Appendix B. Ablation study

### Comparison of Different Threshold Values

The extended ablation in Table 8 evaluates DTC-WSI under a wide range of token retention ratios ($r \in [0.3, 0.8]$) across four benchmark datasets. Performance improves consistently as redundant tokens are removed, with accuracy rising steadily from $r = 0.8$ to $r = 0.5$ on all cohorts. The model achieves its best results at $r = 0.4$, reaching **98.3%** (NSCLC), **97.4%** (BRCA), **96.8%** (RCC), and **94.8%** (PANDA), demonstrating that moderate compression enhances discriminative focus while preserving essential morphology. When compression becomes too aggressive ($r = 0.3$), performance drops sharply—e.g., NSCLC declines from **98.3%** to **90.4%**—indicating loss of critical diagnostic tokens. These

results highlight a clear U-shaped trend: light compression reduces redundancy, moderate compression maximizes accuracy, and over-compression degrades performance. Overall, the study confirms that DTC-WSI benefits most from token retention around $r = 0.4$, where efficiency and predictive power are jointly optimized.

Table 8: Extended ablation study evaluating token retention ratios across four datasets. Metrics reported as Accuracy (Acc) and AUC (%).

| Retention Ratio (r) | TCGA-NSCLC | | TCGA-BRCA | | TCGA-RCC | | PANDA | |
|---|---|---|---|---|---|---|---|---|
| | Acc | AUC | Acc | AUC | Acc | AUC | Acc | AUC |
| r = 0.8 | 95.6 | 96.6 | 94.6 | 95.6 | 93.6 | 94.7 | 92.0 | 93.3 |
| r = 0.7 | 96.1 | 97.2 | 95.1 | 96.1 | 94.3 | 95.4 | 92.8 | 93.9 |
| r = 0.6 | 96.6 | 97.5 | 95.6 | 96.6 | 95.0 | 95.9 | 93.3 | 94.4 |
| r = 0.5 | 97.1 | 98.0 | 96.2 | 97.0 | 95.4 | 96.4 | 93.8 | 94.9 |
| r = 0.4 (Best) | **98.3** | **98.9** | **97.4** | **97.9** | **96.8** | **97.5** | **94.8** | **95.6** |
| r = 0.3 | 90.4 | 92.2 | 93.5 | 94.2 | 88.8 | 89.8 | 85.9 | 86.8 |

## Appendix C. Visualization of Token Compression

We provide visualizations to illustrate how DTC-WSI compresses WSIs while preserving diagnostically important tissue. In Figure. 2, which shows the original WSI of lung adenocarcinoma, the second panel visualizes similarity-guided merging by assigning identical interior and boundary colors to patches that are merged into a single token. This reveals how homogeneous tissue regions—such as smooth stroma or repeated tumor textures—are consolidated into compact groups, while heterogeneous or diagnostically subtle regions remain unmerged. The third panel displays the result of importance-guided pruning, where tokens with low saliency scores are removed entirely, leaving a focused set of highly informative patches concentrated around tumor-rich or otherwise relevant regions. Together, these visualizations demonstrate that DTC-WSI performs structured and interpretable compression, reducing redundancy while retaining the critical morphological patterns needed for accurate WSI classification.

## Appendix D. Algorithm of DTC-WSI

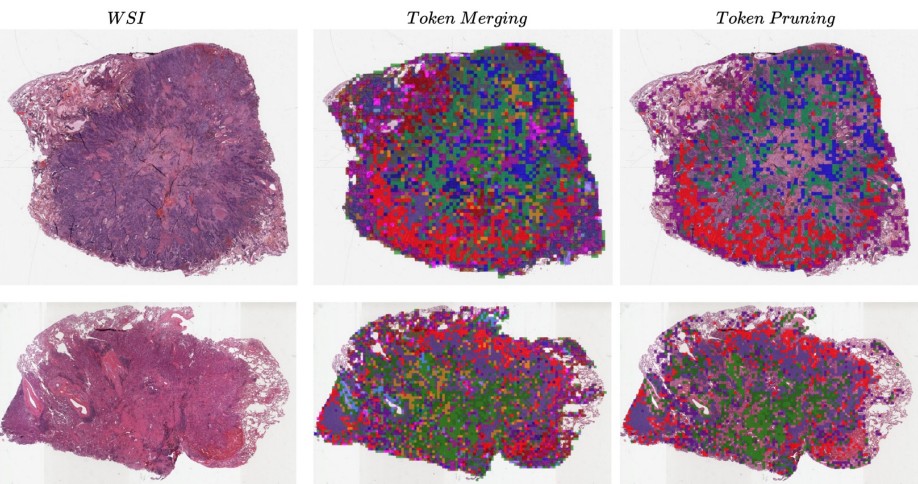

Figure 4: Visualization of the multi-stage token compression in DTC-WSI. (Left) Original WSI thumbnail. (Middle) Similarity-guided merging groups redundant patches into shared representations(Patches with the same inner and border color are merged together.) (Right) Importance-guided pruning removes low-saliency tokens.

**Algorithm 1:** Dynamic Token Compression for Whole-Slide Images (DTC-WSI)

**Input:** Patch features $H^{(0)} = \{h_i^{(0)}\}_{i=1}^{N^{(0)}}$, # stages $T$, target token counts $\{N^{(t)}\}_{t=1}^{T}$, mode
     $\in \{\text{train}, \text{infer}\}$

**Output:** Compressed token set $H^{(T)}$

**for** $t = 0$ **to** $T - 1$ **do**

    $N^{(t)} \leftarrow |H^{(t)}|$ ;                                                `// current #tokens`

    `/* 1.  Importance estimation                                           */`

    **for** $i = 1$ **to** $N^{(t)}$ **do**

        $s_i^{(t)} \leftarrow g_\phi(h_i^{(t)})$ ;                                       `// importance score`

    **end**

    $\alpha^{(t)} \leftarrow \text{softmax}(s^{(t)})$ ;                              `// normalized importance`

    `/* 2.  Bipartite soft matching for token fusion                        */`

    `// Interleaved partition:  odd indices → A, even indices → B`

    $A \leftarrow [1, 3, 5, \ldots], \quad B \leftarrow [2, 4, 6, \ldots]$  Let $L = \min(|A|, |B|)$

    **for** $k = 1$ **to** $L$ **do**

        $i \leftarrow A_k, \quad j \leftarrow B_k \;\; \text{sim}_{ij} \leftarrow \dfrac{\langle h_i^{(t)}, h_j^{(t)} \rangle}{\|h_i^{(t)}\| \, \|h_j^{(t)}\|} \;\; u_{ij}^{(t)} \leftarrow \lambda \, \text{sim}_{ij} - (1 - \lambda) \, |\alpha_i^{(t)} - \alpha_j^{(t)}|$

    **end**

    `// Number of pairs to merge`

    $K^{(t)} \leftarrow \max\big(0, \, N^{(t)} - N^{(t+1)}\big)$  Select top-$N^{(t)}$ pairs $\mathcal{P}^{(t)}$ sorted by $u_{ij}^{(t)}$

    `/* 3.  Merge selected pairs                                            */`

    Initialize $H_{\text{merge}}^{(t+1)} \leftarrow \emptyset$, mark all indices as "unassigned"

    **foreach** $(i, j) \in \mathcal{P}^{(t)}$ *with both* $i, j$ *unassigned* **do**

        $\tilde{h}_i^{(t)} \leftarrow \dfrac{\alpha_i^{(t)} h_i^{(t)} + \alpha_j^{(t)} h_j^{(t)}}{\alpha_i^{(t)} + \alpha_j^{(t)}}$  Add $\tilde{h}_i^{(t)}$ to $H_{\text{merge}}^{(t+1)}$  Mark $i$ and $j$ as "assigned"

    **end**

    `/* 4.  Carry over unmerged tokens                                      */`

    $H_{\text{carry}}^{(t+1)} \leftarrow \{h_k^{(t)} \mid k \text{ unassigned}\} \;\; H_{\text{raw}}^{(t+1)} \leftarrow H_{\text{merge}}^{(t+1)} \cup H_{\text{carry}}^{(t+1)}$

    `/* 5.  Importance-guided pruning                                       */`

    **if** $mode = train$ **then**

        `// soft, differentiable pruning`

        **foreach** $h_k^{(t+1)} \in H_{raw}^{(t+1)}$ **do**

            $m_k^{(t)} \leftarrow \sigma\big(\gamma(\alpha_k^{(t)} - \tau)\big) \;\; h_k^{(t+1)} \leftarrow m_k^{(t)} \, h_k^{(t+1)}$

        **end**

        $H^{(t+1)} \leftarrow H_{\text{raw}}^{(t+1)}$

    **end**

    **else**

        `// hard top-N`$^{(t+1)}$` pruning at inference`

        Rank all $h_k^{(t+1)} \in H_{\text{raw}}^{(t+1)}$ by $\alpha_k^{(t)} \;\; H^{(t+1)} \leftarrow \text{TopK}(H_{\text{raw}}^{(t+1)}, \alpha^{(t)}, N^{(t+1)})$

    **end**

**end**

**return** $H^{(T)}$

