# OpenReview forum: "DTC-WSI: Dynamic Token Compression for Whole Slide Images"
_MIDL.io/2026/Conference — MIDL 2026 Poster_

### Official Review · Reviewer_xfis · 2025-12-23

**Confidence:** 5
**Preliminary Rating:** 3
**Final Rating:** 4

**Summary:**

The authors propose a DTC-WSI method for dynamically compressing tokens to achieve efficient WSI representation. By using a multi-stage token compressor, redundant patches can be eliminated, significantly accelerating inference efficiency and achieving state-of-the-art performance compared to baseline methods.

**Strengths:**

A dynamic token compression method is proposed, which significantly reduces the computational cost of WSI. The compressed tokens can also further improve ABMIL performance in four public WSI datasets.

**Weaknesses:**

- While reducing redundant tokens is indeed a worthwhile approach for WSI, this method should ideally be adapted for Transformer models with quadratic complexity. However, the authors used ABMIL, which is already a lightweight MIL method. I suggest the authors conduct further experiments with some Transformer-based methods [1][2][3] to explore their performance.
- The authors did not compare their method with linear-complexity Mamba-based methods [4][5] to investigate their performance and efficiency.

[1] Ding T, Wagner S J, Song A H, et al. A multimodal whole-slide foundation model for pathology[J]. Nature medicine, 2025: 1-13.

[2] Chen R J, Chen C, Li Y, et al. Scaling vision transformers to gigapixel images via hierarchical self-supervised learning[C]//Proceedings of the IEEE/CVF conference on computer vision and pattern recognition. 2022: 16144-16155.

[3] Wagner S J, Reisenbüchler D, West N P, et al. Transformer-based biomarker prediction from colorectal cancer histology: A large-scale multicentric study[J]. Cancer cell, 2023, 41(9): 1650-1661. e4.

[4] Yang S, Wang Y, Chen H. Mambamil: Enhancing long sequence modeling with sequence reordering in computational pathology[C]//International conference on medical image computing and computer-assisted intervention. Cham: Springer Nature Switzerland, 2024: 296-306.

[5] Zhang J, Nguyen A T, Han X, et al. 2dmamba: Efficient state space model for image representation with applications on giga-pixel whole slide image classification[C]//Proceedings of the Computer Vision and Pattern Recognition Conference. 2025: 3583-3592.

**Detailed Comments:**

- Some statements are incorrect, for example, the TransMIL mentioned in the introduction is not an attention-based MIL.
- There is a lack of ablation studies for $L_{sparse}$.
- Table 2 shows the baseline results for r=0 (i.e., using ABMIL directly without the proposed method), however, the authors state that it requires up to 14.2GB of GPU memory, which contradicts the lightweight efficiency of typical ABMIL. Could the authors explain this?

**Justification Of Final Rating:**

I thank the authors for their efforts and rebuttal. They have addressed my concerns.I also recommend to expand the experiments on other Transformer-based MILs in subsequent phases to enhance the effectiveness of the proposed method. I have changed my score to weak accept.

**Justification Of The Preliminary Rating:**

The comparative baseline is insufficient, the method implementation needs further details, and the authenticity of the presented content needs to be explained. The authors should address these issues.

**Questions To Address In The Rebuttal:**

Please investigate the results of DTC-WSI on transformer-based MIL models and compare them with linear-complexity MIL models like Mamba.  Additionally, provide an explanation of the validity of the model efficiency metrics.

---

> ### Author Response · Authors · 2026-01-25
>
> We really appreciate the reviewer xfis for the thorough review and constructive suggestions. The comments greatly helped us refine the methodology description, strengthen the experimental validation, and improve overall clarity. We respond to each concern below and outline the changes incorporated in the revised version.
>
> **1.	Evaluation with transformer-based and Mamba-based MIL models.**
>
> **Response:** We thank the reviewer for this important suggestion. We extended our experiments by integrating DTC-WSI with transformer-based MIL models and report results for DTC+TransMIL, alongside vanilla transformer MIL baselines. In addition, we compared against linear-complexity Mamba-based methods (e.g., 2dMambaMIL). These results, reported in Supplementary Table 6 and Table-3, show that DTC-WSI consistently improves efficiency and remains competitive or superior in accuracy across different backbones and encoders.
>
> **2.	Incorrect statement regarding TransMIL.**
>
> **Response:** Thank you for pointing this out. We corrected the manuscript to accurately describe TransMIL as a transformer-based MIL method, avoiding the earlier mischaracterization.
>
> **3.	Clarification of ABMIL memory usage.**
>
> **Response:** The reported 14.2 GB GPU memory corresponds to the end-to-end MIL training setup with very large WSI bags (≈10k–20k tokens per slide), batched training, and full backpropagation through all instance embeddings. While ABMIL itself is lightweight, its training-time memory scales linearly with token count and batch size, which becomes substantial in gigapixel WSI settings. To ensure transparency, we revised Table 2 to separately report (i) MIL aggregation-only memory/time and (ii) full-pipeline memory/time, making the efficiency metrics clear and comparable where all results are on our approach with different r valuse.
>
> **4. ablation studies for$\mathcal{L}{\mathrm{sparse}}$ .**
>
> **Response:** We thank the reviewer for pointing this out. We have added an explicit ablation on the sparsity regularization term $\mathcal{L}{\mathrm{sparse}}$ (Table~7). Removing $\mathcal{L}{\mathrm{sparse}}$ leads to denser importance distributions, higher token retention (0.52 vs. 0.40), slower inference (610,ms vs. 410,ms), and degraded performance (96.9% vs. 98.3% accuracy on TCGA-NSCLC). In contrast, the full model with sparsity regularization achieves both better accuracy and stronger compression. These results demonstrate that $\mathcal{L}_{\mathrm{sparse}}$ is a critical component for stabilizing multi-stage token compression and learning compact yet discriminative representations under weak slide-level supervision.

---

### Official Review · Reviewer_jjfC · 2026-01-08

**Confidence:** 3
**Preliminary Rating:** 3
**Final Rating:** 3

**Summary:**

DTC-WSI proposes a token-efficient framework for whole-slide image (WSI) analysis that combines similarity-guided token merging with importance-guided pruning in a progressive multi-stage compression pipeline. The method uses bipartite matching to fuse visually redundant patches and a learned importance network to prune low-saliency tokens, with soft differentiable gates during training and deterministic compression at inference.

**Strengths:**

5.3× speedup and 40% memory reduction with only 40% token retention are substantial and practically meaningful. FLOPs, memory, and inference time are all reported, providing a complete efficiency picture.

Systematic evaluation of retention ratios (r = 0.3 to 0.8) reveals a clear optimal operating point. Ablation isolating merging vs. pruning vs. random selection demonstrates component contributions

**Weaknesses:**

References [1] and [4] are duplicates.
Some figures are low resolution and difficult to interpret.
No comparison to recent token-efficient methods: ToMe, PatchGD, MHIM-MIL (directly applied to pathology)
the pipeline uses a pretrained encoder (CONCH) to produce patch embeddings, but there isn’t an analysis of how sensitive the results are to the choice/strength of that encoder. This makes it harder to isolate whether the accuracy gains come mainly from the proposed compression strategy or from the upstream feature extractor quality.

**Detailed Comments:**

Can you provide standard deviations across 5 folds and significance tests for the claimed 2–4% improvements
Can you add some experiments compared to ToMe,  PatchGD, MHIM-MIL.
Can you clarify: Do all baselines use CONCH features? If not, how much of DTC-WSI's improvement comes from the encoder vs. the compression framework?

**Justification Of Final Rating:**

The authors have carefully addressed all of my concerns in their response. The added clarifications and explanations resolve the previously raised issues and improve the overall clarity of the paper. The response sufficiently strengthens the submission, so I update my evaluation to a weak accept.

**Justification Of The Preliminary Rating:**

DTC-WSI presents a reasonable approach to token-efficient WSI analysis by combining merging and pruning in a principled framework. The efficiency gains are substantial, and the ablations are thorough. However, the methodological novelty is limited, as both bipartite merging and importance-guided pruning are established techniques. The author needs to provide more experiments to show that their modification is important

**Questions To Address In The Rebuttal:**

see above

---

> ### Author Response · Authors · 2026-01-25
>
> We are grateful to the reviewer jjfC for the detailed assessment and valuable feedback. The comments have substantially improved the quality of the manuscript. In the following, we provide point-by-point responses and summarize the revisions implemented in the updated paper.
>
> **1.	Duplicate references and figure quality.**
>
> **Response:** Thank you for pointing this out. We corrected the duplicated references ([1] and [4]) and replaced the low-resolution figures with higher-quality versions, improving readability and clarity throughout the paper.
>
> **2.	Missing comparisons with recent token-efficient methods.**
>
> **Response:** Thank you for this valuable suggestion. We added direct comparisons with recent token-efficient pathology methods, including ToMe, PatchGD, and MHIM-MIL, in Table 1. Under the same experimental protocol, DTC-WSI consistently outperforms these baselines in both predictive performance and computational efficiency.
>
> **3.	Encoder dependence of the results.**
>
> **Response:** To ensure a fair comparison, all main experiments use the same pretrained encoder (CONCH) across all methods. To further assess encoder sensitivity, we added a supplementary ablation (Table 6) evaluating multiple encoders. The results show that DTC-WSI consistently improves performance across different feature extractors, confirming that the observed gains are driven by the proposed compression framework rather than by the choice of encoder.

---

> ### Author Response · Authors · 2026-02-02
> **Update for the final rating**
>
> Hi,
>
> Thank you so much for your thoughtful comments and suggestions — they helped improve our draft. I also appreciate that you updated your rating. It appears your final rating still shows "Borderline"; could you please update it to "Weak accept" when you have a moment?
>
> Thank you again — we appreciate your time and look forward to presenting our work at MIDL 2026.

---

### Official Review · Reviewer_MrFE · 2026-01-08

**Confidence:** 3
**Preliminary Rating:** 2
**Final Rating:** 4

**Summary:**

This paper introduces DTC-WSI (Dynamic Token Compression for Whole Slide Imaging), a framework designed to improve the efficiency and performance of Multiple Instance Learning (MIL) in computational pathology. The method addresses the computational bottleneck of processing gigapixel-scale images by progressively reducing the number of patch tokens through a combination of similarity-guided merging and importance-guided pruning. Experiments conducted on four large-scale datasets (TCGA-NSCLC, BRCA, RCC, and PANDA) demonstrate that the proposed method can achieve 5-10x token reduction and up to 5.3x faster inference while outperforming state-of-the-art MIL baselines in classification accuracy.

**Strengths:**

- Empirical Performance: The method demonstrates a clear reduction in computational overhead (5-10x token reduction) while maintaining or slightly improving AUC/Accuracy on several large-scale WSI benchmarks.
- Implementation Effort: The authors have conducted extensive experiments across four different cancer types, which provides a comprehensive look at how token reduction affects various tissue morphologies.

**Weaknesses:**

- Limited Conceptual Novelty: The proposed "Dynamic Token Compression" appears to be a repackaging of well-known techniques. Token merging is essentially ToMe (ICLR 2023), and token pruning follows the logic of DynamicViT (NeurIPS 2021). The "Dynamic" aspect—making decisions based on input—is a standard feature of these baseline methods. The paper lacks a fundamental theoretical or architectural breakthrough that specifically addresses the unique geometry of WSIs beyond just applying existing CV "tricks."
- Superficial "Dynamic" Mechanism: The "Importance Network" used to guide pruning is conceptually very similar to the attention-based gating mechanisms already prevalent in MIL (e.g., ABMIL, CLAM). Labeling this as a "Dynamic Importance Network" feels like an overstatement of a standard attention-based instance selection process.
- Questionable Efficiency Gains: The paper claims significant speedups, but it does not sufficiently account for the overhead of the Importance Network and the bipartite matching process during training. In a weakly-supervised setting, these extra components might introduce instabilities or hyperparameter sensitivity that are not fully explored.
- Incomplete Comparison: The evaluation focuses on comparing DTC-WSI against standard MIL models (CLAM, TransMIL). However, it fails to compare against other efficient attention mechanisms (e.g., Nyströmformer, Longformer) or simpler sampling-based baselines that could potentially achieve similar efficiency gains with much less architectural complexity.

**Detailed Comments:**

- Hyperparameter Sensitivity: The multi-stage compression introduces several new hyperparameters (e.g., merging ratios and pruning thresholds at each stage). The paper would benefit from a sensitivity analysis showing how these choices affect the final performance.
- Overhead of the Importance Network: While inference is faster, the training process involves bipartite matching and additional network branches. Please provide a table comparing the total training time and GPU memory peak during training versus standard TransMIL.
- Feature Extraction Bottleneck: In WSI pipelines, the bottleneck is often the patch feature extraction (CNN/ViT backbone) rather than the MIL aggregator. The authors should clarify if the reported "5.3x speedup" refers only to the aggregator or the entire end-to-end pipeline.
- Visualization of "What is Merged": While the heatmaps show where tokens are reduced, it would be insightful to show examples of which patches are merged. For instance, does the model correctly merge similar stroma patches while keeping tumor nest patches distinct?
- Ablation on "Dynamic" vs. "Static": A crucial ablation study is missing: comparing the proposed dynamic compression against a "static" version where the same amount of tokens are merged/pruned randomly or based on a fixed grid. This would clarify if the "Importance Network" is truly adding value.

**Justification Of Final Rating:**

Although the methodological novelty is limited (essentially adapting ToMe and DynamicViT to WSI), the rigorous engineering evaluation and the clear efficiency gains (5.3x speedup) make it a valuable contribution for practical pathology pipelines. The logical limitation of the bipartite matching strategy (Set A/B split) is noted, but the empirical results suggest it works well enough in practice.

**Justification Of The Preliminary Rating:**

The paper is primarily an application-oriented work that migrates established token-reduction techniques from general vision transformers to the WSI domain. While the empirical results are positive, the scientific contribution is thin. The "Dynamic" framework feels more like a conceptual rebranding of existing pruning and merging strategies rather than a novel insight into computational pathology. Without a more rigorous comparison against other efficiency-focused architectures or a more unique adaptation to the MIL paradigm, the paper does not meet the novelty threshold for MIDL.

**Questions To Address In The Rebuttal:**

- How does the "Importance Network" differ fundamentally from the attention mechanisms used in ABMIL or CLAM, other than being used for pruning?
- Provide a direct comparison (accuracy vs. FLOPs) with a simple random sampling or top-k attention sampling baseline. Does the "Dynamic" merging actually provide a significant boost over simple selection?
- Why should this be considered a new "Dynamic" framework rather than an application of ToMe/DynamicViT to MIL?

---

> ### Author Response · Authors · 2026-01-25
>
> We sincerely thank the reviewer MrFE for the thorough review and constructive suggestions. The comments greatly helped us refine the methodology description, strengthen the experimental validation, and improve overall clarity. We respond to each concern below and outline the changes incorporated in the revised version.
>
> **1.	Conceptual novelty**
>
> **Response:** We agree that token merging (e.g., ToMe) and token pruning (e.g., DynamicViT) are established techniques. Our contribution is not to introduce new token-reduction primitives, but to propose a WSI- and MIL-specific dynamic compression framework tailored to gigapixel pathology images. Unlike prior methods designed for standard ViTs with a few hundred tokens, DTC-WSI operates on tens of thousands of patch tokens under weak slide-level supervision, where naïve or single-stage compression leads to information collapse. DTC-WSI introduces a multi-stage curriculum compression strategy that jointly combines similarity-guided merging and importance-guided pruning before MIL aggregation, preserving rare diagnostic regions while aggressively removing redundancy. Direct comparisons with ToMe, random/static compression, and pathology-efficient baselines show that these gains cannot be achieved by generic token reduction alone, our approach outperformed over all approaches in Table 1.
>
> **2.	“Dynamic” mechanism and importance network**
>
> **Response:** While attention-based instance weighting is common in MIL, our importance network differs fundamentally in role and effect from ABMIL/CLAM. In ABMIL and CLAM, attention scores are used only for soft weighting during aggregation, and all instances remain in the bag. In contrast, the importance network in DTC-WSI functions as a dynamic computation controller, making input-dependent, stage-wise decisions that actively modify the token set through guided merging and explicit pruning under a fixed token budget. These decisions directly affect which tokens are preserved and how redundancy is fused, not merely how instances are weighted. To avoid overstating novelty, we revised the terminology to describe this module as a saliency-guided compression controller.
>
> **3.	Efficiency gains**
>
> **Response:** We agree that compression overhead and stability must be explicitly accounted for. The reported speedups arise from reducing the token count passed to the MIL/Transformer aggregator, where computation and memory scale with token number (quadratically for transformers). The importance network is a lightweight two-layer MLP with linear complexity, and bipartite matching is implemented using structured pairing, avoiding full pairwise similarity. We added Table 3, reporting training time per epoch, peak GPU memory during training, and end-to-end inference time (including compression overhead), showing that the overhead is modest relative to the downstream savings. We also added a hyperparameter sensitivity analysis, demonstrating stable performance across a broad operating range.
>
> **4.	Incomplete comparison**
>
> **Response:**  We expanded the evaluation to include simple sampling-based baselines (random sampling and top-k attention sampling) and efficient long-sequence models used in pathology (ToMe, PatchGD, MHIM-MIL, Longformer, and 2DMambaMIL). All methods are evaluated under matched token budgets and identical encoders. Results show that while simpler methods reduce computation, they consistently underperform DTC-WSI, confirming the benefit of dynamic, saliency-aware merging and pruning. These results are now reported in the main table (Table 1) and supplementary material.
>
> **Additional clarifications and analyses**
>
> •	**Hyperparameter sensitivity:** Added analysis over retention ratios in supplementary Table 5 showing robust performance without fine-tuning.
>
> •	**Training overhead:** Added efficiency comparisons against standard TransMIL, showing modest overhead relative to savings in Table 3.
>
> •	**End-to-end vs. aggregator-only speedup:** Clarified that the reported 5.3× speedup refers to the MIL aggregation stage, and additionally reported full end-to-end runtime including compression overhead in Table 2.
>
> •	**Visualization of merged tokens:** Added qualitative visualizations showing that DTC-WSI merges redundant regions (e.g., stroma/background) while preserving tumor structures in Figure 3.
>
> •	**Dynamic vs. static ablation:** Added an ablation comparing dynamic compression to static/random sampling under the same token budget (in  Table 4), demonstrating consistent gains from input-adaptive compression.

---

> > ### Comment · Reviewer_MrFE · 2026-01-28
> >
> > I thank the authors for their detailed response. The inclusion of the missing baselines and efficiency analysis (Table 3) is appreciated. However, significant issues regarding presentation clarity and rigor remain:
> >
> > 1. Poor Documentation of Experimental Setup
> > It was unnecessarily difficult to verify the validity of the main results. The specific feature extractor (CONCH) used for the primary comparisons (Table 1 and Table 3) is not explicitly stated in the main captions or the corresponding experimental setup text. I had to search through the Ablation Study (Table 6) to eventually deduce that the high metrics were achieved using CONCH. Critical details that define the baseline performance should be transparently presented rather than buried in supplementary ablation tables.
> >
> > 2. Critical Errors in Visualization (Figure 3)
> > The newly added Figure 3 contains a confusing and contradictory error. The caption states that "red boxes" represent "pruned patches", but the very next sentence claims that "high-importance patches retained" are also in "red boxes". Visually, the patches in the red boxes (right column) appear to be largely background/glass. This aligns with the logic of "pruning" but completely contradicts the text claiming they capture "diagnostically relevant tumor tissue patterns." Presenting background noise as tumor tissue, even if it is just a labeling error, is a significant oversight in quality control.

---

> > ### Author Response · Authors · 2026-01-28
> >
> > We thank Reviewer MrFE for the careful follow-up and for highlighting these important presentation issues. We fully agree with these concerns and addressed both points. Also, we uploaded a newly revised version of the manuscript incorporating the corresponding corrections, as detailed below.
> >
> > **(1) Clarification of experimental setup and feature extractor**
> >
> > We agree that the feature extractor used for the main comparisons should be explicitly stated. Although we currently mention the use of CONCH as the feature encoder in Section 2.2, we acknowledge that this information should be more explicitly stated. In the final version, we revised the captions of Tables 1 and Table 3, as well as the performance comparison section, to clearly specify that CONCH is used as the default feature encoder for all primary results, including both accuracy and efficiency comparisons.
> >
> > **(2) Correction and clarification of Figure~3 visualization**
> >
> > We sincerely apologize for the confusion caused by the labeling inconsistency in Figure~3. This was a presentation error. In the final version, we revised the figure caption as follows:
> >
> > **Figure~3: Visualization of multi-stage token compression in DTC-WSI across retention ratios $r \in {1.0,,0.7,,0.5,,0.4}$. Example merged patches (e.g., adipose or stroma) are shown in blue boxes, pruned patches (e.g., background or slide borders) in red boxes, and high-importance patches retained for final prediction (e.g., tumor regions) in green boxes, highlighting diagnostically relevant tissue patterns.**
> >
> > In the previous caption, diagnostically relevant patches (which are in a green box) were mistakenly indicated using the same color as pruned regions (red box), which led to the apparent contradiction. We revised the caption, and the updated figure will clearly demonstrate that DTC-WSI suppresses background artifacts while preserving tumor-associated regions.
> >
> > We appreciate the reviewer’s careful attention to these issues, which will help improve the clarity and overall quality of the final manuscript.

---

> > > ### Comment · Reviewer_MrFE · 2026-01-31
> > >
> > > I thank the authors for their prompt response and for addressing the remaining presentation issues. The explicit confirmation of the experimental setup (specifically the use of the CONCH encoder) ensures that the primary results in Tables 1 and 3 are now properly contextualized for reproducibility, and the correction to the Figure 3 caption effectively resolves the visual contradiction regarding pruned versus retained tokens. These revisions significantly improve the clarity and rigor of the manuscript.

---

### Author Rebuttal · Authors · 2026-01-25

**Rebuttal:**

We revised the manuscript in response to the reviewers’ comments, added additional experiments, and provided point-by-point responses for each reviewer (please find the file in supporting material). We sincerely appreciate their careful reviews and constructive feedback, which have significantly improved the quality of the manuscript.

**Supporting Material:**

/attachment/70a1cd44e088c351bc68aaa30519aab8520a791f.pdf

---

### Meta-Review · Area_Chair_GKaH · 2026-02-03

**Recommendation:** Accept (Poster)
**Confidence:** 4

**Metareview:**

All reviewers' final scores are weak accept (although jjfC did not actually change their final rating to WA despite saying so in the final justification). MrFE highlighted the evaluations of the method's practicality, jjfC stated that their previously raised issues were resolved via clarifications and explanations, and xfis stated that their concerns were addressed by the rebuttal. The remaining concerns and suggestions do not disqualify the paper from acceptance in my opinion, so I recommend acceptance.

---

### Decision · Program_Chairs · 2026-02-13

Accept (Poster)